# Non-Wood Forest Products' Marketing: Applying a S.A.V.E. Approach for Establishing Their Marketing Mix in Greek Local Mountain Communities

**Marios Trigkas** [1,*], **Foteini Pelekani** [2], **Ioannis Papadopoulos** [1], **Dimitra C. Lazaridou** [3] and **Glykeria Karagouni** [1]

1  Department of Forestry, Wood Sciences and Design, University of Thessaly, 43100 Karditsa, Greece; papadio@uth.gr (I.P.); karagg@uth.gr (G.K.)
2  Env-Consults, 42132 Trikala, Greece; fpelekani@uth.gr
3  Department of Forestry and Natural Environment Management, Agricultural University of Athens, 36100 Karpenisi, Greece; dlazaridou@aua.gr
*  Correspondence: mtrigkas@uth.gr

**Abstract:** The contribution of non-wood forest products is especially important in the context of rural sustainable development. Nevertheless, their perceived economic and environmental value remains low. The lack of an explicit and effective marketing strategy for NWFPs tailored to local mountain communities' needs may lead to their restricted access to the market and underestimation of their value. The aim of this paper is to gain knowledge regarding the components of a marketing mix that could support the local markets of NWFPs, in Greek mountainous areas. The paper presents an analysis of the marketing mix for NWFPs, following for the first time the S.A.V.E. approach. The research contributes to the existing literature as we seek to "meet" groups of NWFPs' consumers, in order to develop a customer-centric value proposition in Greek local mountain communities. The findings indicate that the marketing and promotion of NWFPs requires not only knowledge of the relative products and the market, it requires analysis and knowledge of the specific needs of local mountain communities and the ways that needs are met by the attributes and characteristics of the NWFPs as part of the solution that they can offer. Local mountain communities in Greece, through the exploitation of NWFPs, are trying to face challenges regarding the improvement of their income and their general wellbeing level. Also, we propose as a part of the marketing mix for NWFPs, their promotion as products with a distinct spatial, local identity, by associating them with local "culture economies". Finally, we argue that a customer-centric marketing mix of NWFPs, which focuses on customers' needs, desires, and resources as the starting point of the planning process, involves a higher level of mixing and synergies creation along the whole value chain, than simple personalization, with customers to interact with suppliers using ICT and by personal time disposition connected to nature.

**Keywords:** non-wood forest products; marketing; SAVE marketing mix; NWFPs value; local mountain communities

## 1. Introduction

Non-wood forest products (NWFPs) are defined as goods derived from forests that are tangible and natural objects of biological origin other than wood [1]. They contribute to various economic sectors, such as rural development and environmental regulation, and play a significant role in sustainable forest management and bioeconomy [1–4]. NWFPs in the Mediterranean region are an important source of income [5,6], providing important recreational and commercial activities in the rural–forest regions. Their economic value is also recognized in many countries across Europe, as the use of NWFPs is of great importance in the countries of Central and Eastern Europe [5,7–9]. Recent research indicates a widespread impact of NWFPs on the rural–urban relationship [10], with most NWFPs

providing additional sources of income while they are at the heart of nature-based solutions and many European policy ambitions [6,11]. Forest management planning and the development of criteria related to NWFP's treatment in forest management plans are essential to properly harness multifunctional synergies [12]. Besides their popularity in certain regions, their value seems to be underestimated when they are confronted as marketable goods and research is rather limited in this area.

Similarly, in Greece, the importance of NWFPs has been under-recognized, whereas little attention has been paid to the development of an integrated marketing strategy for them. On the other hand, there is a wide variation in the extent to which NWFPs are used from region to region and even between households within a community. Local trade of NWFPs exists and it has advantages to offer; moreover, local markets can provide a guaranteed way of reaching consumers and play a critical role in enhancing livelihoods and improving income opportunities. In many cases, local markets tend to be informal supply chains with the same individuals performing all functions along the supply chain. However, such products hardly figure in the statistics since their production–consumption cycle is limited within local barriers; therefore, NWFPs deserve more attention from both research and business development perspectives. Indicatively, research on NWFPs' markets should also be pointed towards exploring pathways for increasing the relative returns for participants and/or pave the way for more people to engage [7,13].

Our research purports to focus on the above need by establishing a basis for a targeted and tailored marketing mix for NWFPs to become facilitators for the sustainable development of local communities and mountain areas. This is in line with the relevant literature; in sustainable forestry, marketing has been often treated as an effective tool of the function among forest resources management, processing, and end-use [14]. The use of the S.A.V.E. marketing mix model ensures the attention, and especially the involvement of consumers, in order to develop an effective value proposition for NWFPs in Greek local mountain communities.

A major limitation is the fact that different categories of NWFPs require different marketing strategies, making it difficult to establish rules that can be applied to the entire range of NWFPs [15]. However, an initial bottom-up approach for the creation of marketing strategies, combined with effective policy interventions and sustainable forest management strategies, may encourage local people to become further and deeper involved in exploitation of non-wood forest resources [16,17]. From the perspective for livelihood, the marketing of NWFPs is defined as an increase in their value in trade, which is expected to increase income and employment opportunities, especially for the poor and other less-favored rural communities [18–21]. This is a focal point of the present research. It is important to identify ways to improve the employment and income generation potential of NWFPs, through practices related to the transition towards a forest-based circular bioeconomy and marketing, which drives demand for NWFPs [2,10,22,23]. Uncovering the weaknesses of NWFPs as territorial products and identifying a potential effective marketing mix, our research could contribute to the creation of opportunities, especially for new sustainable entrepreneurial ventures at the local level, resistible to market pressures, and adjustable to customer demands [24,25]. On the other hand, local marketing strategies using contemporary implementation measures are powerful tools that can be used to increase both the production and marketing of NWFPs [17,24,26,27].

Several studies have been conducted internationally on the issue of developing relevant strategies in different contexts [21,24,28]; however, most of them are not customer-centric [29–31] and focus on traditional marketing approaches. Besides the relevant research, this seems to also be a common fact for the Greek entrepreneurial reality as well, since many Greek micro-businesses often get locked in the old-fashioned business models, neglecting the logical or even emotional needs of their customers [32–34]. This way of thinking and acting becomes even more emphatic regarding NWFPs, as it is a rather traditional sector that concerns the local economies of mountainous areas. The present research addresses this need to use a marketing mix that interprets the dynamics of modern customer-centric

business models and confronts NWFPs as potential business ideas that promise viable businesses. This is why the S.A.V.E. model (solution, access, value, education/engagement) was used [35]. The model promotes a more complex approach in the modern marketplace. It allows for better adaptation to customer needs by giving weight to contemporary market challenges and generally following a more customer-centric approach. This model structure enables the creation of customers who are committed around the brand of a product or business at any given time through the provision of a narrowed value and participatory content of that value in alignment with customer needs and behavior [35,36].

Thus, the aim of this study is to gain knowledge regarding the core components of a marketing mix that have an influence on making NWFPs a viable business in Greek mountainous areas. The model will demonstrate the strengths and main barriers for the engagement in NWFPs production and certain prerequisites, such as the creation of a strong brand for Greek NWFPs. Through the study, it is also investigated if marketing and branding techniques can help to establish a market for NWFPs in Greek mountainous areas based on synergies among stakeholders and the integration of NWFPs' value chain. Following a bottom-up approach, our effort is to identify how the utilization of NWFPs will be able to contribute to addressing the needs of local mountain communities. In this context, the approach is based on the application of a new, customer-centric model for defining the marketing mix of NWFPs, targeting the formation of a coherent value proposition, in response to both local communities and the contribution of NWFPs to the multifunctional management of forest ecosystems. According to our knowledge, this is the very first attempt to apply the S.A.V.E. marketing mix model to forestry production and especially to NWFPs. It is also the first attempt of using this approach in relation to mountain communities and exploring how to engage them in a contemporary marketing strategy for NWFPs.

## 2. Methodology

### 2.1. The S.A.V.E. Marketing Mix Model

Following the logic of the S.A.V.E. marketing mix model, we seek to "meet" groups of NWFPs' consumers, through access to multiple channels of communication, promotion and distribution, asking for their attention and especially their involvement, in order to develop a value proposition for NWFPs in Greek local mountain communities The goal is therefore to engage with consumers in a two-way, creative and effective communication relationship, always in their own terms of engagement. Thus, the following approach was developed as opposed to the traditional 4 ps model of the marketing mix [35]:

- Instead of the product, we focused on the solution. Our effort was to define the attributes of NWFPs in terms of covered needs/solutions of local Greek mountain communities;
- Instead of place, we focused on access by exploring ways to create an integrated system of NWFPs' collectors', traders' and retailers' presence based on the overall "buying journey" of the customer, rather than focusing on individual distribution points and communication channels;
- Instead of price, we tried to determine the value of NWFPs. Thus, our survey referred to the benefits of NWFPs that can be expected by local mountain communities. Our effort was to link the value of NWFPs to the problems and address the needs that can result from offering related products in local markets;
- Finally, instead of promotion, the survey explores ways of consumer engagement in the local mountain markets of NWFPs, by providing information relevant to differentiated needs at each point in the customer's life cycle. This approach is used as opposed to the one-way information of traditional communication specialties, because modern marketing, seeks two-way engagement, by participating and responding with ideas, suggestions, judgments, and criticisms from customers.

### 2.2. Data Collection

The current survey was implemented from November 2022 to January 2023 among 473 individuals who live in mountainous areas. Data were collected through a questionnaire distributed online and through in-person interviews among people living in mountain areas. The use of both online and paper versions of the questionnaire allowed us to increase the sample size. Regarding the online respondents, they were asked firstly to state if they reside in a rural region and to specify it. This enabled us to focus on the answers of those reside in Greek mountain areas. Data were merged to a dataset to be analyzed. The questionnaire includes 5 different sections and 18 questions in total. The first section includes questions that refer to socio-demographic characteristics of the respondents, such as age, educational level, occupation, and income to establish their profile. The following sections focus on questions related to respondents' views on the characteristics of NWFP's, their access to NWFP's markets, the value created by them, and activation and engagement in the related market. All questions were close-ended and most of them should be answered using the 7-point Likert scale (with values from 1 to 7). The main research was preceded by a pilot to test the reliability and validity of the questionnaire. The development of scales was based on a relevant literature review, as well as empirical and theoretical contributions from NWFP's scholars in Greece [11].

### 2.3. Sampling Method

Based on the data provided by the National Statistical Service of Greece, the mountainous population of the country was calculated to 828,259 residents, according to the census of 2011. To calculate the sample size, Slovin's Formula for finite population $n = N/(1 + Ne^2)$ was used with a confidence level set at 95% and an alpha level at 0.05, where: $n$ = sample size and $N$ = population (828,259) [37,38]. Thus, the minimum sample size needed to our survey, based on an acceptable margin of error was 400 respondents. The sample finally consisted of 473 respondents, which is quite acceptable.

### 2.4. Data Analysis

Cronbach's alpha test ($\alpha$) was applied to determine the internal consistency among Likert-type items. Data, which reached an acceptable level of reliability, or were correlations emerged among the items, were further subjected to principal component analysis (PCA) [39]. Principal components analysis with varimax rotation was used to analyze the results of the Likert-scale items related to the formation of the S.A.V.E. factors of the NWFPs marketing mix, namely, solution, access, value, and engagement for local mountain communities. It was applied to the multivariates with the aim of reducing the number of the initial variables to fewer composite variables. The PCA transforms the original variables into new axes, so that the data presented in those axes are uncorrelated with each other [40].

Using factor analysis, we tested if the items of the questionnaire statistically belonged to the dimensions, i.e., the factors used to describe the different S.A.V.E. factors of the NWFPs marketing mix. Eigenvalues were used to ensure that each item was distributed with a high load to each of the main factors [39,40]. Factor loadings also showed the correlation of each item with a main factor that emerged from the analysis. Items with high factor loadings (over 0.3) in the rotation component matrix were selected as the main ones that significantly contributed to the description and determination of the main S.A.V.E. factors that emerged [39]. All analyses were performed using Statistical Package for Social Sciences (SPSS) version 27.0.

## 3. Results

The socio-demographic characteristics of survey participants are reported in Table 1. The gender distribution was relatively evenly split between male (47.4%) and female (52.6%). In terms of age, substantial shares of participants were aged between 45 and 54 (28.8%), between 18 and 24 (19%), between 25 and 34 (16.5%), and between 55 and 64 (16.1%). Nearly

40 percent of participants reported a monthly household income between EUR 601 and EUR 1200, with 28.9% under EUR 600 and 23.3% between EUR 1201 and 1800. Regarding the educational level, considerable percentages of respondents had completed primary school (24.3%). Nearly 20% of the people reporting their education level had completed secondary school (20.5%) and 35% high school. Lastly, 20 percent of respondents had a bachelor's degree. The composition of the survey sample can be considered sufficiently representative of the Greek mountainous population, although the potential selection bias is an issue in an online survey as the respondents choose freely to participate or not.

**Table 1.** Demographic characteristics of participants.

| Variable | Categories | Percentage (%) |
|---|---|---|
| Gender | Male | 47.4 |
| | Female | 52.6 |
| Age | 18–24 | 19.0 |
| | 25–34 | 16.5 |
| | 35–44 | 15.4 |
| | 45–54 | 28.8 |
| | 55–64 | 16.1 |
| | 65+ | 4.2 |
| Income | Under EUR 600 | 28.9 |
| | EUR 601–1200 | 37.2 |
| | EUR 1201–1800 | 23.3 |
| | EUR 1801–2400 | 6.6 |
| | EUR 2400–3000 | 1.8 |
| | EUR 3000 or more | 2.3 |
| Education | Primary school | 24.3 |
| | Secondary school | 20.5 |
| | High school | 35.0 |
| | Bachelor's degree | 20.2 |

The initial results relate to participants' answers to a set of probing questions focusing on specific aspects of the process of retrieving responses, such as understanding and recalling information for the NWFPs, in order to explore how they interpret the concepts analyzed in the rest of the questionnaire. Thus, to capture the meaning that the respondents attribute to the NWFPs, all respondents were asked firstly to select those definitions that best suit them. Results show a clear domination of two definitions "All forest products except wood" and "Forest products harvested directly from the forest other than wood", with percentage 66.4% and 59.2%, respectively. As NWFPs' definitions are diverged, some respondents considered the NWFP as "Animal products" (36.2%), as "Forest products grown within the forest, and forest areas other than wood" (28.8%). NWFPs have been also defined according to their growing method, in particular "Forest products that can be produced by conventional agricultural methods" was selected by 13.5% participants and "Forest products grown outside forest and woodland other than wood" by 12.3%. Respondents were asked about their interest in NWFPs and their willingness to buy them, as well as about their perceptions on NWFPs' attributes. The overall estimate of the reliability of the questions was good (Cronbach's a = 0.744), indicating an overall interest for most of the NWFP. The most preferred product among respondents was honey (mean = 4.58 ± 0.741), while nuts (mean = 4.43 ± 0.789), aromatic plants (mean = 4.43 ± 0.849), herbs (mean = 4.42 ± 0.837), and forest fruits (mean = 4.13 ± 0.930), were among the predominant preferences as well. The above results further confirm the validity of the survey, as the participants seem to have a good level of awareness and information about NWFPs, and they show interest for NWFPs' markets in their areas of residence.

### 3.1. Solution by NWFPs at Local Markets

Our analysis revealed that the estimated reliability coefficient for sub-questions regarding NWFPs' attributes were not acceptable (Cronbach's a = 0.372); however, it is interesting that respondents assign their attributes related to "Nutritional value" (mean = 4.32 ± 0.724), "Organic origin" (mean = 4.12 ± 0.843), and "Better taste compared to the cultivated ones" (mean = 4.08 ± 0.925). Moreover, respondents assign NWFPs' attributes related to higher aesthetic value, but also to higher selling price (mean = 3.26 ± 1.249 and mean = 3.64 ± 0.953, respectively). The specific results reveal that the above factors correlate to each other. For that reason, we also applied a correlation analysis between the attributes of the NWFPs, which resulted to the correlation of the following:

- Organic origin with nutritional value, better taste, and aesthetic value (Pcc = 0.503, 0.453 and 0.129, respectively, with *p* < 0.01);
- Nutritional value with better taste and aesthetic value (Pcc = 0.449, 0.149, respectively, with *p* < 0.01);
- Better taste with aesthetic value (Pcc = 0.281 with *p* < 0.01).

Thus, we further elaborated our data using PCA. The analysis showed that the following two key factors accounted for the NWFPs attributes (Table 2):

1. Attributes that cover needs to local communities' wellbeing;
2. Attributes that cover needs for enhancement of local income.

**Table 2.** Principal component analysis (PCA) with varimax rotation on NWFPs' attributes.

| Variable/Attributes | Mean | SD | Component | |
|---|---|---|---|---|
| | | | PC1 | PC2 |
| Better taste compared to the cultivated ones | 4.08 | 0.925 | 0.794 | |
| Nutritional value | 4.32 | 0.724 | 0.783 | |
| Organic origin | 4.12 | 0.843 | 0.780 | |
| Aesthetic value | 3.26 | 1.249 | 0.429 | |
| Higher selling price | 3.64 | 0.953 | | 0.911 |
| Eigenvalue | | | 2.043 | 1.659 |
| % of variance | | | 34.056 | 27.648 |

Our results give a first direction towards the offered solutions that NWFPs have to contribute in terms of covered needs of local Greek mountain communities.

### 3.2. Access to NWFPs' Markets

Among the important findings of this research is the highlighting of local markets in mountainous areas as disposition channels. It is interesting to observe that the overwhelming majority of respondents (99.1%) believe that local markets in mountainous areas could contribute to enhancing the sales of NWFPs. According to most of the respondents (93.02%) stated that locals engaged in NWFPs' collection should also be the ones to consider better ways of promotion. However, 85.17% of the respondents attribute the responsibility for the promotion of NWFPs to local authorities, such as municipalities, and 59.83% of them to forestry services. Furthermore, according to the results, rural entrepreneurship seems to be quite important too; NWFPs' processing enterprises (78.44%), tourism sector enterprises (77.38%), and cultural associations (76.11%) are indicated by respondents to play a significant role too. However, the rather poor Cronbach's alpha coefficient (0.592) indicates that the respondents do not have a clear knowledge of the NWFPs, or even their value and the potential ways of enhancing it. This is a significant hint that non-wood forest products' marketing was never considered in Greek mountainous areas. Residents were asked to evaluate the distribution channels that facilitate consumer's access to NWFPs' markets. The analysis extracted three components, which account for 60.36% of the total variance. This percentage was deemed to be adequate because, in the social sciences, it is not uncommon to regard a solution that accounts for 60% of the total variance as

satisfactory [41]. As can be seen in Table 3, about 34% of the total variation is explained by the first principal component (PC1), 16.9% by the second principal component (PC2), and 9.4% by the third principal component (PC3). In other words, 60.36% of the total variance in the 13 considered variables can be condensed into three new variables (PCs). As in Table 2, the three components are arranged in an order of reduced variances. The first component PC1 represents "Specialist selling", and consists of the variables "Special events (e.g., festivals)", "Agritourism", "Cultural associations", "Physical points of sale", "Specialized points of sale", "Complementary products in physical store", and "Marketed as a superfood". Component PC2 represents "Retail and wholesale market", and consists of the variables "Retail market", "Wholesale market", and "Other". The third component PC3 represents "Direct collection and distribution through internet", and consists of the variables "Direct collection of NWFPs", "Websites and e-shops", and "Social media".

**Table 3.** Principal component analysis (PCA) with varimax rotation on NWFPs' distribution channels.

| Variable/Distribution Channels | Mean | SD | Component | | |
|---|---|---|---|---|---|
| | | | PC1 | PC2 | PC3 |
| Special events (e.g., festivals) | 4.38 | 0.752 | 0.815 | | |
| Agritourism | 4.33 | 0.759 | 0.760 | | |
| Cultural associations | 4.07 | 0.896 | 0.737 | | |
| Physical points of sale | 4.33 | 0.664 | 0.714 | | |
| Specialized points of sale (hotels, restaurants, cafes, etc.) | 3.88 | 0.882 | 0.697 | | |
| Complementary products in physical store | 3.92 | 0.811 | 0.650 | | |
| Marketed as a superfood | 3.87 | 1.030 | 0.555 | | |
| Retail market | 3.11 | 1.272 | | 0.851 | |
| Wholesale market | 2.95 | 1.211 | | 0.849 | |
| Other | 2.39 | 1.172 | | 0.701 | |
| Direct collection of NWFPs | 3.35 | 1.077 | | | 0.734 |
| Internet (websites and e-shops) | 3.98 | 0.852 | | | 0.702 |
| Social media | 3.81 | 0.938 | 0.437 | | 0.695 |
| Eigenvalue | | | 4.430 | 2.193 | 1.225 |
| % of variance | | | 34.073 | 16.871 | 9.424 |

According to the results, "specialist selling" is mostly represented by cultural events and places and tourism; this indicates that NWFPs are still confronted as "complementary products" and not as a significant niche market, as they should be. These results support and complement the results regarding access to NWFPs' markets. In general, access to the NWFPs' markets can be through the following:

1. Experiential buying channels linked to consumer needs and local cultural and leisure markets;
2. Established distribution channels;
3. A personal engagement and market research based on technology and personal time disposition connected to nature.

In addition, residents were asked how often they see NWFPs advertisement on internet. The analysis of the findings shows that 54.8% of the respondents "Rarely" meet NWFPs web advertisements, whereas 18% "Seldom". This fact indicates that online advertising for NWFPs is a product that has not fully developed yet, and maybe more effective online advertising strategies need to be established.

### 3.3. Value of NWFPs

The value of NWFPs was investigated from both an economic and an environmental perspective. Cronbach's alpha was used to evaluate the internal reliability of the sub-questions assessed for the overall contribution of NWFPs (Table 4). The reliability statistic is considered very satisfactory, as Cronbach's alpha index scored $\alpha = 0.873$. According to

respondents' perceptions, the economic contribution of NWFPs to the income of the population living close to forests receives the highest evaluation scores (Mean = 4.43 ± 0.674). From the responses, the significant contribution of NWFPs in "Creating more successful tourism products" (Mean = 4.36 ± 0.659) and in "Preservation of local cultural heritage" (Mean = 4.19 ± 0.730) is highlighted. The contribution of NWFPs to "Sustainable development" (Mean = 4.03 ± 0.705) and the "Provision of bio-based raw materials to industrial sectors (e.g., pharmaceutical industry)" (Mean = 4.02 ± 0.790) is also highly estimated. The contribution of NWFPs to forest protection and sustainable management (Mean = 3.99 ± 0.894, ± 0.766) are equally evaluated by interviewees, whereas NWFPs contribution to "Biodiversity conservation" and "Curbing climate change" scored lower (Mean = 3.78 ± 0.841 and Mean = 3.53 ± 0.846, respectively).

**Table 4.** Cronbach's alpha coefficient of responses regarding the economic and environmental contribution of NWFPs.

| Variable | Mean | SD | Reliability Analysis | | |
|---|---|---|---|---|---|
| | | | Squared Multiple Correlation | Cronbach's Alpha If Item Deleted | Cronbach's Alpha |
| | | | | | 0.873 |
| Enhancement the income for population living close to forest | 4.43 | 0.674 | 0.508 | 0.862 | |
| Creating tourism product | 4.36 | 0.659 | 0.492 | 0.864 | |
| Preservation of local cultural heritage | 4.19 | 0.730 | 0.523 | 0.855 | |
| Health and prosperity | 4.18 | 0.734 | 0.461 | 0.857 | |
| Sustainable development | 4.03 | 0.705 | 0.467 | 0.856 | |
| Provision of bio-based raw materials to industry | 4.02 | 0.790 | 0.410 | 0.861 | |
| Forest protection | 3.99 | 0.864 | 0.451 | 0.858 | |
| Sustainable management | 3.99 | 0.766 | 0.517 | 0.854 | |
| Biodiversity conservation | 3.78 | 0.841 | 0.411 | 0.866 | |
| Curbing climate change | 3.53 | 0.846 | 0.323 | 0.869 | |

The application of PCA (Table 5) highlighted two key factors regarding the value proposition that could be included in the marketing mix and the strategy for the promotion of NWFPs in the local markets of mountainous Greece. As mentioned, we refer to the benefits of NWFPs that can be expected by local mountain communities in our effort to link this value to addressing the needs in local markets. Additionally, correlating these key factors with the results that emerged from the solution part of the NWFPs' marketing mix (Table 2), we can support that the key factors of the value proposition for the NWFPs in local Greek mountain communities stand as follows:

1. NWFPs can contribute to local income enhancement based on cultural, tourist, and bioeconomy markets under the principles of sustainability;
2. NWFPs can contribute to local mountain communities' wellbeing through the productive, ecological, biospheric, and social services and amenities they provide.

**Table 5.** Principal component analysis (PCA) with varimax rotation on NWFPs' value proposition.

| Variable/Distribution Channels | Mean | SD | Component | |
|---|---|---|---|---|
| | | | PC1 | PC2 |
| Enhancement the income for population living close to forest | 4.43 | 0.674 | 0.855 | |
| Creating tourism product | 4.36 | 0.659 | 0.800 | |
| Preservation of local cultural heritage | 4.19 | 0.730 | 0.642 | 0.420 |
| Provision of bio-based raw materials to industry | 4.02 | 0.790 | 0.633 | 0.301 |
| Health and prosperity | 4.18 | 0.734 | 0.626 | 0.387 |
| Forest protection | 3.99 | 0.864 | 0.486 | 0.524 |
| Sustainable development | 4.03 | 0.705 | 0.433 | 0.623 |

**Table 5.** *Cont.*

| Variable/Distribution Channels | Mean | SD | Component | |
|---|---|---|---|---|
| | | | PC1 | PC2 |
| Sustainable management | 3.99 | 0.766 | 0.396 | 0.676 |
| Biodiversity conservation | 3.78 | 0.841 | | 0.814 |
| Curbing climate change | 3.53 | 0.846 | | 0.764 |
| Eigenvalue | | | 4.740 | 1.227 |
| % of variance | | | 47.398 | 12.270 |

*3.4. Engagement in NWFPs' Local Markets*

A core point of the present research is the engagement of consumers. In this vein, potential ways of engaging consumers in the local mountain markets of NWFPs were investigated. Cronbach's alpha coefficient indicated good internal consistency for the overall scale (Cronbach's a = 0.88) for the items used. Respondents believe that the market penetration of NWFPs could be enhanced through the provision of training programs (Mean = 4.28 ± 0.691), even if this training is conducted at a university level (Mean = 4.25 ± 0.707). Moreover, participants suggest that setting criteria for organic certification of NWFPs could help in this direction, as well (Mean = 4.20 ± 0.699). The support of any kind of promotional campaign is highly evaluated by the respondents. They regard those campaigns supported by local authorities (Mean = 4.10 ± 0.756), by ministries (Mean = 4.02 ± 0.778), or by enterprises (Mean = 3.97 ± 0.756), which could highlight the value of NWFPs. The inclusion of NWFPs in the national policy and the setting of criteria in forest management plans regarding the exploitation of NWFPs are equally highly rated by the respondents (Mean = 4.05 ± 0.793).

In the same vein, the PCA revealed the following two key factors that could support the customer engagement dimension of the NWFPs' marketing mix model for local mountain communities in Greece:

1. Engagement should be ensured through supporting a two-way customer relation, service provision to and from businesses, and policy measures for innovation support for NWFPs' market, which responds to contemporary forestry challenges (Eigenvalue = 6.244, % of variance = 44.602);
2. Engagement by other approaches (Eigenvalue = 1.110, % of variance = 7.927).

At this point, it is worth mentioning that all items used to describe ways of engagement, constituted reliable and significant elements in the PCA as formulated one main key factor, while the item "other" stranded as the second factor alone. However, no significant and clear answers were given to this question by the respondents, except the following that "it would be useful to link the NWFPs' market with the Mediterranean diet and cultural heritage" showing an interesting approach focusing on social and cultural characteristics and services of NWFPs.

The final S.A.V.E. marketing mix model for NWFPs in Greek local mountain communities that emerged is as follows (Table 6).

**Table 6.** The existing S.A.V.E. marketing mix model for Greek local communities on NWFPs.

| Marketing Mix | Factors |
|---|---|
| Solutions | • NWFPs cover needs to local communities' wellbeing<br>• NWFPs cover needs for enhancement of local income |

**Table 6.** *Cont.*

| Marketing Mix | Factors |
| --- | --- |
| Access | • Experiential buying channels, linked to consumer needs and local cultural and leisure markets.<br>• Through established distribution channels<br>• Personal engagement and market research, based on technology and personal time disposition connected to nature |
| Value | • Contribution to local income enhancement, based on cultural, tourist and bioeconomy markets, under the principles of sustainability<br>• Contribution to local mountain communities' wellbeing through the productive, ecological, biospheric, social services and amenities they provide |
| Engangement/Education | • Supporting a two-way customer relation, service provision to/from businesses, and policy measures for innovation support for NWFPs' market, which responds to contemporary forestry challenges |

## 4. Discussion

Non-wood forest products are a multifaceted part of cultural heritage of Europe and Greece particularly, constituting an integral part of everyday life, especially for local mountain communities. They contribute to the achievement of the United Nations Sustainable Development Goals, in particular, the social and cultural dimension, the environmental dimension, and the economic dimension. As such, NWFPs are a potential source of nature-based solutions that can make a significant contribution to Europe's policy priorities under the European Green Deal [41], but they face risks and threats. These are dictated by both global and local challenges, such as climate and land use change, uncontrolled harvesting, poor management, and illegal trade, together with difficult market competition with similar products based on fossil or non-renewable resources. These risks are exacerbated by a lack of systematic knowledge of, inter alia, the levels of natural resource availability (distribution, productivity), harvesting and cultivation techniques, domestication, as well as official, reliable data on production, consumption, and trade, along with appropriate labeling and quality standards, ultimately affecting the implementation of effective marketing strategies as well [6].

On the other hand, the number of stakeholders directly or indirectly involved in the NWFPs' value chains is even larger, as it also includes those who collect for personal consumption, as well as non-professional collectors. This happens not only in the local mountain communities around the world, but it is a phenomenon which characterizes the global markets of NWFPs in Greece as well. The limited bargaining capacity of producers and collectors, the existence of non-renewable substitutes and imported near-equivalents, combined with a manufacturing sector consisting mainly of small- and medium-sized enterprises (SMEs) with low innovation, creates high competitiveness in markets, limited profit for producers and collectors at the bottom of the chains, and, consequently, a high risk of abandonment of the activity, which is particularly noticeable in local communities of higher income countries [42,43]). There is a need to reverse this situation, to recognize the positive social and environmental externalities of NWFPs' value chains and to reduce competition with non-renewable or, sometimes, cultivated counterparts. The present survey supports the need for scientific research adapted to the specificities of each region and the needs of each social group, for the crafting of personalized and targeted marketing strategies for NWFPs. Obviously, there is no universal solution, so holistic and well-tailored marketing strategies are needed. It is necessary to integrate the NWFPs into the economic trend of the market and to exploit their potential for sustainable spatial development. Immediate mobilization is required to increase revenues for the landowners, not only to

ensure access to the resource for amateur collectors, but also to enable a fair income and adequate working conditions for practitioners, as well as jobs and livelihoods for local communities in a broader context [6].

We argue that our research focuses on the above needs to create a targeted and tailored marketing mix for NWFPs to become facilitators for sustainable development for local communities and mountain areas. The research highlights both the importance of the NWFPs and their inadequate marketing. It appears that NWFPs, in the mind of the consumers, constitute territorial goods with a rather complementary economic role and contribution to rural income. They are mostly connected to recreational and cultural activities, and local points of sales at village squares and roads where collectors go to sell their everyday collections. Even when they consider enterprises, they regard mostly cultural and touristic ones. Furthermore, NWFPs do not seem to have crossed the local boarders yet, while advertisement is rare according to the research. The findings indicate the clear need to change this perspective of the consumers and build new niche markets for the various categories of the NWFPs.

A novel customer-centric marketing mix should emphasize the value proposition of NWFPs, which should be developed as actual healthy, organic, and high-value products. It should also support enlargement from "Specialist selling" to retail and internet market while enhancing further the links between NWFPs and culture, territorial marketing, and sustainable tourism. Therefore, a deeper analysis is needed in order to provide concrete suggestions for specific marketing mix models for specific NWFPs' categories.

After exploring the factors that can contribute to the establishment of NWFPs' marketing mix in Greek local mountain communities, it appears that we have to rethink their value and develop new niche markets. This involves the creation of a strong brand for Greek NWFPs, and the redesign of the whole value chains, building models on the specific needs of the local mountain communities and, at the same time, going international with the use of technology. This imposes the need for certification of primary and processed products through appropriate action plans. Educating consumers appeared to be a significant factor in order to highlight the value of NWFPs and, at the same time, engage them in the goods of nature. Although the results indicate training, education could be also performed through social media and sites.

The results highlighted that local mountain communities in Greece, through the exploitation of NWFPs, are trying to face challenges regarding the improvement of their income and their general wellbeing level, which is in accordance with the related literature [6,7,20,42,44–46]. Livelihoods in mountain areas are considerably more susceptible to environmental and economic changes because of the rough topography, remoteness, and poor socioeconomic infrastructure. On the other hand, markets for NWFPs niche products are growing and this is a significant opportunity for locals. We argue that NWFPs can open new global markets, continue to be important territory products in the local markets of mountain communities, while, at the same time, creating new jobs and increasing the income of the inhabitants, preserving the cultural and natural heritage of the area. Values such as the contribution of NWFPs to improving health and wellbeing as they supply bio-based raw materials to important industrial sectors, and the ecosystem services provided by NWFPs are not negligible [3,4,47]. The above can work both ways, resulting in incentives for the creation of new businesses, enhancing the potential for circular economic activities in the mountainous regions while contributing to a low environmental footprint. Their promotion as commercial products of added value will have multiple positive effects by attracting new investors to mountainous areas for ventures of innovative businesses, having the NWFPs as a reference point.

NWFPs could reach the final consumer through specific access channels adapted to create an integrated system of NWFPs' overall "buying journey" of the customer, rather than focusing on individual distribution points and communication channels. The results show that a possible part of the marketing mix in the context of the wider marketing strategy for NWFPs is to promote products with a distinct spatial, local, or regional identity. By

associating products with "culture economies" or local images, such as cultural traditions and heritage, the value of the product is increased because consumers associate specific regions with specific products. The shift towards quality food has provided significant opportunities for doing business in a new economic environment more able to cope with the forces of globalization [6]. This has also resulted to the present research as part of the engagement elements that could be used in the NWFPs' marketing mix model for local communities.

Although not a part of the primary research, secondary research regarding NWFPs' markets in Greece revealed that complex rules and regulations that bar mountain communities managing small entrepreneurial schemes for NWFPs and enjoying the benefits of this growing market. NWFPs hold special values and have niche markets. Enabling policies and supporting rules and regulations for marketing strategies of mountain products can benefit mountain regions and population and help them obtain value for their products and efforts. Distinctive features and values of NWFPs can be highlighted in novel ways such as through green certification and ecolabeling, access and benefit sharing initiatives, green marketing [17], and fair trade as part of the effort to link the value of NWFPs to the problems and addressing the needs that can result from offering related products in local markets [3,48,49].

Furthermore, the parallel use of opportunities provided by information and communication technologies will gradually narrow the boundaries of local markets for NWFPs and expose economic activity to greater external competition. The limited scale and scope of local markets of NWFPs force rural entrepreneurs to develop innovative products and efficient marketing to compete with their counterparts in urban areas. On the contrary, regions that fail to participate in the adoption and growth of these technologies risk marginalization. As digital marketing continues to grow, it will prove to be a powerful tool that will improve sales, display information about the business and NWFPs/services, and build customer relationships more effectively and efficiently [50,51]. Thus, local entrepreneurs and customers will enjoy mutually rewarding relationships based on the development of interactivity [52–54]. Effective advertising will be that which is presented in an interactive media environment, as customers have a much more active role in acquiring information and interacting online by gaining access to the final purchase. In addition, our findings highlight that a customer-centric marketing mix of NWFPs, which focuses on customers' needs, desires, and resources as the starting point of the planning process, involves a higher level of mixing than simple personalization [54,55]. Customers need to interact with suppliers and by personal time disposition connected to nature as part of the access to the NWFPs. Of course, the use of established access channels has also been highlighted.

As already discussed, the distinctive features and values of NWFPs can be highlighted as part of the effort to link the value proposition of NWFPs to the problems and address the needs of local mountain communities and markets of Greece. In conclusion, we could, therefore, suggest that it is of great importance to support local value chains and local networks for NWFPs. The shortening of NWFPs' value chains fosters closer links between consumers, local producers, entrepreneurs, and traders, often allowing for further "space" for improvement [6,23]. The sizes of local supply and demand, available local capacities, and good practices are the preconditions for success factors [28,47,56]. The regional and national authorities of Greece should support existing initiatives or seek new opportunities to develop local value chains by promoting new business models and vertical integration models [57].

## 5. Conclusions

While, in general, NWFPs are highly valued, their great potential is not always fully recognized, especially in terms of the improvement of local income and wellbeing. In fact, the quantities produced, the geographical areas, different qualities, consumption, trade patterns, and the state of conservation are—with few exceptions—the usual barriers to the development of their marketing strategy. This knowledge gap explains their inadequate

treatment in the context of policies and an often-cited mismatch between sectoral policies and needs or opportunities from investors, which hampers public, collective, and private action, especially in local mountain communities. The lack of effective and coordinated marketing strategies hampers the potential contribution of NWFPs to addressing societal challenges and the creation of sustainable livelihoods. This is a challenge for the Implementation of sustainable and multifunctional management of forest ecosystems. Highlighting the value of NWFPs, in terms of their potential to improve the income and wellbeing of local people, can provide a strong incentive for the development and implementation of management measures focusing on improving the processes of production, harvesting, and finally, marketing of NWFPs. The present research is original for the Greek context, as it is the first time that market research for NWFPs is carried out to draw useful conclusions aiming at formulating a marketing mix for this specific market of forest products in Greece as a lever to support local economies based on contemporary customer-centric approaches. Thus, it is the first time that the model of the S.A.V.E. approach is used in the marketing mix of NWFPs. This model structure provides an opportunity to create customers who are committed around a product's particular brand at any given time through the delivery of targeted value and participatory content of that value in alignment with customer needs and behavior. The research outlined the existing factors of NWFP's marketing mix, which can be used as a first step to reform; this means the creation of niche markets for the Greek NWFPS, building their brand based on the added value they deliver to local communities, opening new channels, and creating synergies.

**Author Contributions:** Conceptualization, M.T. and F.P.; methodology, M.T.; software, M.T. and D.C.L.; validation, M.T. and F.P.; formal analysis, M.T.; investigation, F.P.; resources, F.P.; data curation, M.T. and D.C.L.; writing—original draft preparation, M.T. and F.P.; writing—review and editing, M.T., G.K. and I.P.; visualization, M.T.; supervision, M.T. All authors have read and agreed to the published version of the manuscript.

**Funding:** This research received no external funding.

**Acknowledgments:** This manuscript is a revisited version of the original one under the title "Marketing of Non-Wood Forest Products. Defining their marketing mix in local mountain communities of Greece", submitted to the 11th International Conference on Contemporary Marketing Issues which took place between the 12t and 14th of July 2023 at Corfu Island, Greece.

**Conflicts of Interest:** The authors declare no conflict of interest.

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
