# Peer review of "Non-Wood Forest Products’ Marketing: Applying a S.A.V.E. Approach for Establishing Their Marketing Mix in Greek Local Mountain Communities"

_forests, doi:10.3390/f14091762_

Round 1

Reviewer 1 Report

The search for additional sources of income for forest management is one of the priorities listed by the European Union in the New Forest Strategy, as well as an action indicated by social certification systems for sustainable forest management. The use of non-wood forest products (NWFPs) has therefore become particularly important in recent years. As authors mention, “Estimates show that in many southern and eastern countries, the value o fNWFPs, far exceeds that of timber”. However, this branch of forestry it is still underestimated.

The use of NWFPs should also be seen as an extremely important element of regional development, especially (but not only!) in developing countries. There are not many examples of forest management focused on non-timber benefits. In this context, the importance of market regulation and marketing policy of NWFPs should be emphasized. The authors are aware of this problem, e.g. writing in the introduction: “In many cases local markets tend to be informal supply chains”. The development of the use of NWFPs must be supported by reliable scientific research, including taking into account the need to promote and support the development of small enterprises whose operation is based on the acquisition, processing and distribution of NWFPs and, above all, marketing policy. ​In this context, the issues raised in the reviewed article seem to be extremely important.

The article solves the research problem methodically, based on a rich literature reviev.

In the “Introduction” chapter (by the way, excellent and comprehensive), the authors write: “NWFPs, particularly in the Mediterranean region, are an important source of income (...) providing important recreational and commercial activities in the rural-forest regions.” This sentence is simply not true! It is well known that in Europe, the use of NWFPs is of greatest importance (albeit informally!) in the countries of Central and Eastern Europe (Latvia, Lithuania, Poland, the Czech Republic, Slovakia, Ukraine,...) - this is also demonstrated by, among others, in one of the publications cited by the authors: Lovrić, M., Da Re, R., Vidale, E., Prokofieva, I., Wong, J., Pettenella, D., ... & Mavsar, R. (2021 ). Collection and consumption of non-wood forest products in Europe. Forestry: An International Journal of Forest Research, 94(5), 757-770. Although the reviewed article concerns Greece, at least one sentence in the above context should be included in this chapter.

The results are very valuable, clearly presented and carefully analysed.

According to the reviewer, a particularly valuable chapter is "Discussion". It is very extensive, logically composed, and above all - it shows the complexity of non-timber forest use and highlights the challenges related to the future of this branch of forestry. Just one minor note: In this chapter the authors write: “On the other hand, the number of stakeholders directly or indirectly involved in the NWFPs’ value chains is even larger, as it also includes those who collect for personal consumption, as well as non-professional collectors. This happens mainly in the local mountain communities around the world, as in Greece as well”. The reviewer has only one remark to this sentence: this applies not only to mountainous areas/communities.

The conclusions are accurate, resulting from the assumed aim of the research. One of them in particular deserves to be highlighted: “The lack of effective and coordinated marketing strategies hampers the potential contribution of NWFPs to addressing societal challenges and the creation of sustainable livelihoods.” This sentence is worth treating as a challenge for sustainable forest management.

Reviewer 2 Report

Advantages of the paper:

In this paper, the S.A.V.E. method model is applied to analyze the marketing mix of non-timber forest products in Greek mountain areas, and the attribute, economic and environmental contribution of non-timber forest products, and distribution channels of Non-Wood Forest Products market are analyzed by PCA method, which is innovative to a certain extent. Finally, it also gives specific suggestions on how to do the marketing and promotion of non-timber forest products.

Disadvantages of the paper:

1. In the introduction part, it explains the research content of this paper, but in the last part of the introduction, it is only a short sentence to explain the innovation of this paper, and it should be expanded appropriately; At the same time, this paper does not reflect the necessity of research, that is, why does this paper need someone to study? It needs to be explained in the article.

2. The chapter distribution of the full text lacks rationality, the number of words in the introduction is too much, which is close to 30% of the whole article (excluding references), and the language expression needs to be further condensed.

3. In the introduction, this paper mentioned that Non-Wood Forest Products can bring economic and environmental benefits to forest areas, and pointed out the practical significance of the study. However, the whole article does not show the theoretical significance of the research and is not convincing.

4. As for the sample survey, there is no descriptive statistics on the survey sample results, which is simply expressed in words and should be displayed in charts to judge whether the data is reasonable and effective.

5. There are redundant words in this paper. The last two sentences of the first paragraph beginning with result 3 describe the results of the survey in words, but do not reflect the significance or function of this paragraph for this paper. Why should it be displayed?

6. When Cronbach's alpha coefficient was not met in parts 3.1 and 3.2 of this paper, there was no clear explanation as to why the analysis could still be carried out. At the same time, the cumulative variance contribution rate of common factors should be more than 80%, but it is far lower than this value in this paper, so there is reason to doubt whether PCA method can be applied.

Minor editing of English language required

Round 2

Reviewer 2 Report

 The paper has been revised and improved to some extent, but in the introduction, the importance of the research in the article, the reasons for reaching such a conclusion, and the possible reasons can be written in more detail.

Minor editing of English language required,Especially in the abstract, conclusion, and discussion section.
